# Exploring In-Context Learning for Knowledge Grounded Dialog Generation

**Qinyu Chen, Wenhao Wu and Sujian Li**
School of Computer Science, Peking University
National Key Laboratory for Multimedia Information Processing, Peking University
{chenqinyu,waynewu,lisujian}@pku.edu.cn

## Abstract

Large neural-based dialog generation models have been applied in many real-life scenarios, yet they are prone to hallucination and tend to produce factually inaccurate outputs which raise great concerns. To alleviate this problem, we propose a plug-and-play retrieval-based framework **IKA**, which leverages in-context learning and retrieval techniques to enhance LLMs on knowledge grounded dialog generation. We design thorough experiments on a large-scale knowledge graph with 1M+ facts (Moon et al., 2019) to investigate the effectiveness and generalization of our framework. Experiments show that our method surpasses previous training-based SOTA by a large margin, specifically **46.67%** in BLEU4, **26.01%** in ROUGE-L, **122.90%** in BARTScore and **30.50%** in Entity Coverage F1. Further analysis shows promising abilities of LLMs to perform knowledge-intensive tasks, which is previously considered weak and understudied.

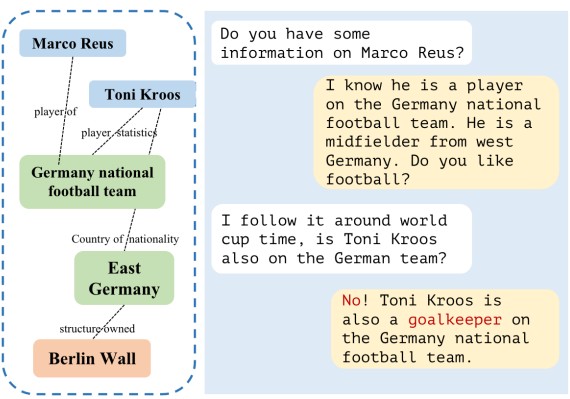

Figure 1: An example of conversation wherein the agent receives *correct* knowledge but still generates *hallucinated* response. Toni Kroos is the midfielder rather than the goalkeeper of Germany national football team, which contradicts the knowledge paths provided.

## 1 Introduction

Existing knowledge grounded dialogue (KGD) systems aim to make use of external knowledge to produce reasonable dialogue responses (Ji et al., 2022; Dziri et al., 2021; Zhang et al., 2019; He et al., 2017). To introduce the KGD task, Figure 1 gives an example; the left side of the figure provides the corresponding knowledge graph, while the right side of the figure shows the dialogue history and the response. The goal of KGD is to generate fluent and faithful response based on context and knowledge.

The earlier works of KGD focus on the use of knowledge graphs, where sub-graphs are retrieved using entities and relationships given dialog history and relevant knowledge. Generally, they utilize a separate knowledge encoder module. Errors in knowledge retrieval stage may propagate to the generative model (Sarkar et al., 2022; Ji et al., 2022; Kang et al., 2022). This type of methods aims to

compensate for the lack of relevant knowledge in the generative model, which are usually inefficient in training and unsatisfactory in performance.

Recently, large language models (Brown et al., 2020; OpenAI, 2023) have been applied to a variety of tasks. They store rich parametric world knowledge through pre-training and are able to produce fluent and human-like responses. However, they uncontrollably generate content that is inconsistent with factual knowledge and even contain hallucinations (Shuster et al., 2021; Dziri et al., 2022b,a; Mielke et al., 2022). Thus, hallucination in large language models (Ji et al., 2023) is a big concern in real-world applications, as a growing number of users directly integrate LLMs into their daily pipelines without realizing this risk (Tiku, 2022). We also use GPT3.5(text-davinci-003) to conduct preliminary experiments on KGD, where the context and knowledge are concatenated as textual input to generate response, and find similar

hallucination problem. We can see the ability to leverage knowledge in LLMs is non-trivial. How to leverage knowledge is the focus of our research.

Large language models like GPT-3 (Brown et al., 2020) and LaMDA (Thoppilan et al., 2022) have demonstrated emergent in-context learning abilities, which becomes a new research paradigm for LLMs. In this paradigm, task definition, demonstration examples and target context are concatenated as the input prompt for LLMs. The example selection process is vital to the effectiveness of ICL (Min et al., 2022) We believe a more refined example **retrieval** stage based on both dialogue history and knowledge is needed. To the best of our knowledge, no previous work (Ji et al., 2022; Kang et al., 2022; Dziri et al., 2021) has explored the potential of ICL in knowledge grounded dialogue generation.

In order to overcome these issues, we propose **IKA** framework as an effective plug-and-play solution. The principle of IKA is to retrieve high quality in-context demonstrations to guide LLMs. A high-level overview is illustrated in Figure 2. IKA consists of three modules, a retrieval module for retrieving relevant examples based on history and knowledge, a DER module to improve the diversity of retrieved examples and a prompt constructing module to build the final input for LLMs. By plug-and-play it means no additional training or parameter updates are required. The retrieval module can be implemented by any retrieval methods, and the inference LLMs can be either open-source or black-box.

Experimental results show that our method significantly outperforms state-of-the-art (Ji et al., 2022) on OpendialKG(Moon et al., 2019). In automated evaluation, our framework is able to gain **46.67%** on BLEU4, **26.01%** on ROUGE-L, **122.90%** on BARTScore and **30.50%** on Entity Coverage F1. For the choice of retrieval method, our best results have **18.43%**, **27.75%** and **69.78%** relative gains on BLUE, ROUGE-L and BARTScore (Yuan et al., 2021) compared to random retrieval baseline. Our proposed diversify strategy achieves the highest faithfulness evaluation results. The quantitative and qualitative investigation further demonstrates its efficacy. In summary, our contributions are as follows:

- We are the first to apply in-context learning to knowledge grounded dialog generation. Ablation experiments demonstrate the outstanding performance of ICL for knowledge-intensive task.

- We give a comprehensive empirical study on many aspects of ICL strategies in KGD, including the choice of retrieval methods, knowledge representation etc. These findings are valuable for a deeper understanding of LLMs.

- We propose a plug-and-play retrieval based framework called **IKA**(**I**n-context **K**nowledge grounded dialog **A**ugmenter). IKA is applicable across LLMs and significantly surpasses previous works (Ji et al., 2022; Sarkar et al., 2022; Dziri et al., 2021).

## 2 Related Works

### 2.1 Hallucination in Open-domain Dialogue Generation

How to reduce the false information in generated text is a popular research topic in natural language generation. Previous works have studied the hallucination in sub-field of NLG such as abstractive summarization or machine translation (Huang et al., 2021; Lee et al., 2018). Dialogue generation is different from above tasks in that it requires multi-turn, long-term interaction. Dziri et al. (2022b) propose to split hallucination in open-domain dialogue into intrinsic and extrinsic hallucination. Some methods propose to improve the transformer (Vaswani et al., 2017) architecture to increase dialog faithfulness. Xu et al. (2021) leverage light-weight adapter as knowledge expert to enhance GPT-2 (Radford et al., 2019). Wu et al. (2021) apply an inductive attention mechanism for self-attention-based generation models. Instead of improving the generation model itself, Dziri et al. (2021); Ji et al. (2022); Sarkar et al. (2022) propose to add extra module such as knowledge graph grounding module or re-ranking module to reduce hallucination.

### 2.2 In-context Learning

Large language models pre-trained on next-token prediction objective have been found to have the ability to adapt to new task with only a few examples as input (Brown et al., 2020), which means they can be applied to different tasks at inference time without expensive training. However, the theory and mechanism of in-context learning have not been well studied. Some researchers (Wang et al., 2023) argue that in-context learning happens when

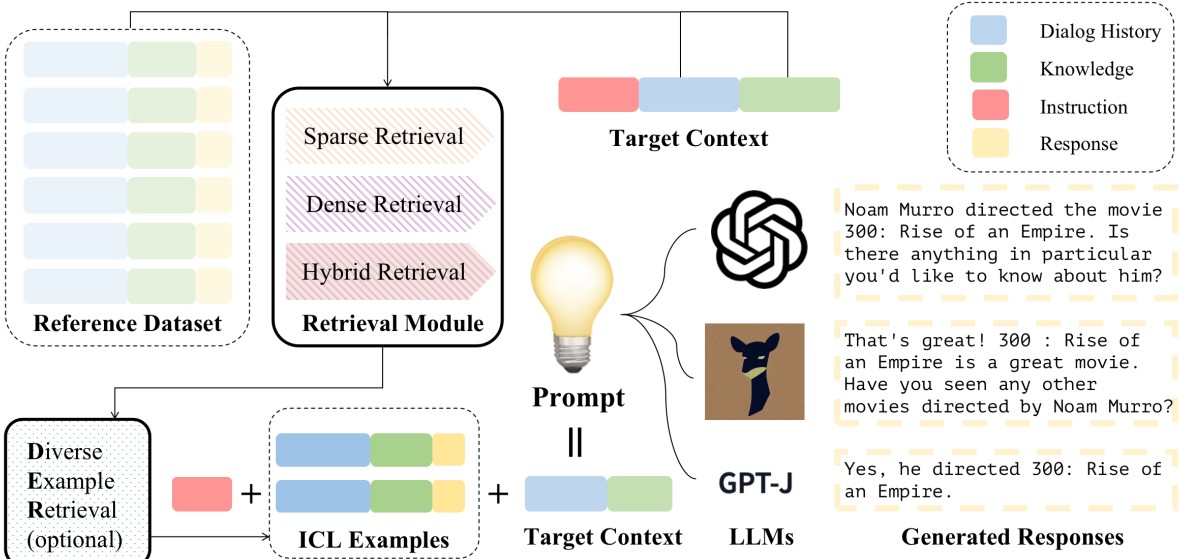

Figure 2: An overview of IKA framework.

a latent concept is inferred at test time. Olsson et al. (2022) contribute in-context learning ability of transformer-based model to the existence of "induction heads". Akyürek et al. (2022) further verify this hypothesis on toy linear regression. The example selection process has also become a popular research topic. Liu et al. (2021) propose to retrieve in-context examples using nearest neighbor algorithm. In supervised learning, the input-label correspondence is very important, but Min et al. (2022) argue that ground truth labels are not required in ICL, which is quite counter-intuitive.

## 3 Methodology

We propose a plug-and-play retrieval based framework called **IKA**(**I**n-context **K**nowledge grounded dialog **A**ugmenter) to enhance LLMs' faithful generation ability. The overview of our method is presented in Figure 2. We conduct broad experiments on mainstream LLMs such as GPT-2 (Radford et al., 2019), GPT-J (Wang and Komatsuzaki, 2021), GPT-3 (Brown et al., 2020), GPT-4 (OpenAI, 2023) and Falcon (Almazrouei et al., 2023). Most of them are general purpose LLMs pretrained on large-scale corpus and have not been fine-tuned or optimized for KGD task.

We find KGD especially suitable for in-context learning mainly for two reasons: (1). the provided knowledge is very helpful in selecting semantically similar demonstrations (2). how to leverage given knowledge and generate high quality

response is complex even for powerful LLMs. Recent work (Evanson et al., 2023) points out the similarity between training large language models and children's language acquisition, therefore we argue that similar mechanism may exist in prompting language models to perform specific task. An analogy is instructing a high school student to pass openbook exam. A smart teaching technique would be to provide some well selected input-output examples so that the student understands how to use given knowledge to produce high quality answers.

In this section, we first give a formal description of KGD task and introduce current research challenges. Then, we formulate KGD task under a conditional text generation framework using incontext learning techniques. In addition to traditional in-context examples selection strategies, we design several retrieval strategies to ground our selection process on dialogue history and relevant knowledge.

### 3.1 Task Definition

For dialogue systems, generating responses involves handling data samples that include a dialogue history $H$, a collection of utterances $U$ that reflect interaction between human users and AI assistants.

Here we focus on knowledge grounded dialog task, which is an extended setting of dialogue response generation. In KGD task, the conversation system is typically equipped with a knowledge base that contains a wide range of information in the

form of text or triplets. Factual triplets are often presented in the form of *(subject, predicate, object)*. The subject and object are entities in the knowledge graph, and predicate represents the relationship between the two. The goal of KGD is to generate faithful and informative response based on dialog history $H$ and knowledge $K$. We have faithfully followed the setup of previous works ([Zhou et al., 2021](); [Dziri et al., 2021](); [Ji et al., 2022]()).

## 3.2 Model KGD as Conditional Text Generation via In-Context Learining

We propose to model KGD as a conditional text generation ([Hu and Li, 2021]()) problem. The output sequences are conditioned on the combination of instruction, in-context examples, dialog history and knowledge. To the best of our knowledge, we are the first to model KGD under the framework of in-context learning. Concretely, the probability of generating a target $y$ given input $x$ can be modeled as :

$$\mathcal{P}_{LM}(y|x) = \prod_{t=0}^{n} \mathcal{P}_{LM}(y_t|H, K, y_{<t}) \quad (1)$$

where *LM* denotes the parameters of language models, *H* denotes dialog history and *K* denotes knowledge. In the setting of in-context learning, examples are required to instruct LLMs, thus the above equation can be rewritten in the following form:

$$\mathcal{P}_{LM}(y_t|H, K, y_{<t}) = \\ \sum_{\mathcal{C}} \mathcal{P}_{LM}(y_t|H, K, \mathcal{C}, y_{<t})\mathcal{P}_r(\mathcal{C}|H, K) \quad (2)$$

where *C* denotes the retrieved in-context examples given history and knowledge. Here we decompose the conditional text generation probability into the product of two terms. $\mathcal{P}_r(C|H, K)$ corresponds to the joint probability distribution of in-context examples where $r$ denotes the example retrieval process. However, depending on different retrieval strategies, the joint probability distribution of examples can be modeled differently. For instance, we can randomly sample examples from reference dataset, the joint distribution of examples and the distribution of history and knowledge are independent, so $\mathcal{P}_r(C|H, K) = P_r(C)$. Previous work ([Liu et al., 2021]()) shows that random sampling leads to unstable performance, and further argues to select examples that are semantically-similar to the target context, i.e. calculating $\mathcal{P}_r(C|H, K)$. In this work,

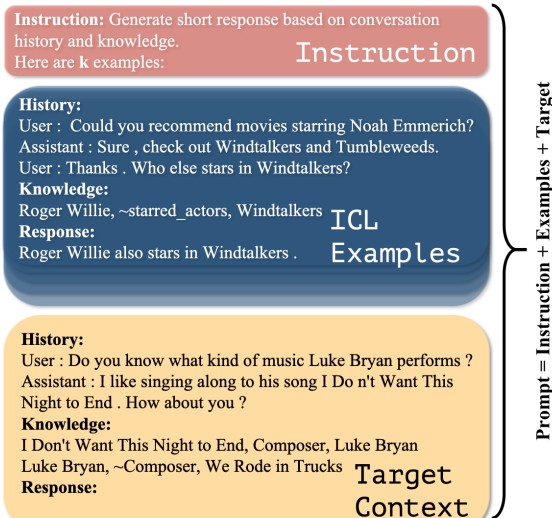

Figure 3: An example of prompt

we propose to model the dependency between examples, which can be expressed as:

$$\mathcal{P}_r(C|H, K) = \prod_{i=1}^{n} \mathcal{P}_r(C_i|H, K, C_{<i}). \quad (3)$$

This equation implies the dependencies in the process of example selection, which means that the examples selected first may influence the selection process afterwards. However, In practice, due to the high computational cost, it is not necessary to model the full dependency. Empirical results show that it is more efficient to model example retrieval as a Markov process, where the selection of current example only depends on target context and previously selected example:

$$\mathcal{P}_r(C|H, K) = \prod_{i=1}^{n} \mathcal{P}_r(C_i|H, K, C_{i-1}). \quad (4)$$

After examples are retrieved, we optionally re-rank them using similarity metrics, more details can be found in next subsection. Last, $k$ demonstrations are placed in descending order according to their retrieval score and the target context is appended at the end to form the final prompt that is fed to LLMs. A prompt example is illustrated in Figure 3.

## 3.3 Diversified Example Retrieval Strategy

As mentioned earlier, the selection of examples is very important, and we need to ensure both the fidelity of the selected examples and a certain amount of diversity. In recommender systems,

there is a common task of multi-objective recommendation, such as ensuring both the relevance and the diversity of the retrieval. We drew inspiration from this (Sá et al., 2022) and design the **Diverse Example Retrieval** (DER) algorithm.

First, we retrieve $N$ relevant examples $C_{ini} = \{C_1, C_2, \ldots, C_N\}(N \gg n)$ based on query context $q$, where $q$ is the combo of dialog history $H$ and knowledge $K$. Notice that the initial $K$ examples can be retrieved by any retrieval method, our goal is to find a diverse subset $C_{div} = \{C_1, C_2, \ldots, C_n\}$. Next we find the example with the highest retrieval score and put it into a queue. We calculate the distance between the remaining examples and the example at the end of the current queue, and get the diverse score $S$ according to Eq.(5). The final score is a linear combination of distance score and retrieval score controlled by hyper-parameter $w$:

$$\mathcal{S}_{diverse}(\mathcal{C}_i) = (1-w) * distance(\mathcal{C}_i, \mathcal{C}_{last}) \\ + w * s^{ret}(\mathcal{C}_i, q) \quad (5)$$

where $q$ denotes the query context, $s^{ret}$ the retrieval score and $C_{last}$ the last example in the queue. At last, we put the example with the highest diverse score at the end of queue. We repeat this process until there are $n$ examples in the queue. In this way, the selected examples are both relevant to the input and distinct from the ones that have already been chosen. Algorithm 1 is a pseudo-code description of **DER**.

---

**Algorithm 1:** Diverse Example Retrieval

**Data:** Initial Retrieved Examples
    $C_{ini} = \{C_1, C_2, \ldots, C_N\}$(N$\gg$ n),
    Empty Queue $Q$, Query Context **q**,
    Example Number $n$
**Result:** Diverse In-context Examples:
    $C_{div} = \{C_1, C_2, \ldots, C_n\}$

1 **start**;
2 Find example $C_1$ with the highest retrieval score: $Q.push\_back(C_1), C = C/C_1$;
3 **while** $length(Q) < k$ **do**
4     $C_{temp} \leftarrow Q[-1]$;
5     $S_{div} \leftarrow dist(C_{temp}, C/C_{temp})$;
6     $S_{sim} \leftarrow sim(\mathbf{q}, C/C_{temp})$;
7     $S \leftarrow (1-w) * S_{div} + w * S_{sim}$;
8     $r \leftarrow \arg\max S$;
9     $Q.push\_back(C_r), C = C/C_r$;
10 **end**
11 **return** $Q$;

---

## 4 Experimental Setup

We mainly experimented on **OpendialKG** using different retrieval strategies and different numbers of in-context examples. To ensure the universality of our method, we first run experiments on OpenAI's GPT-3.5 (text-davinci-003) via APIs to verify the effectiveness and set hyper-parameters. Then we test our method on several LLMs. We set temperature to 0 in all our experiments to avoid the impact of randomness.

### 4.1 Dataset & Preprocessing

**OpendialKG** is currently the only public open-domain English dialog dataset that is annotated with **knowledge graph path** according to Yu et al. (2022). It also provides the textual form of knowledge corresponding to each knowledge triplet. A series of previous works have validated the effectiveness on this dataset (Ji et al., 2022; Dziri et al., 2021; Sarkar et al., 2022), so we choose to experiment on this dataset as well. We also experimented on **WOW**(Dinan et al., 2018) which only contains textual knowledge and yields similar results. Previous works filter OpendialKG by keeping dialogue samples that are annotated with at least one knowledge path. We do additional filtering to ensure that at least one knowledge path is actually **used** in the generated responses. We hope to select demonstrations that genuinely reflect how knowledge is leveraged and thus can serve as better examples.

### 4.2 Language Models

We explore the effectiveness of proposed framework primarily on GPT-3.5(text-davinci-003) (Ouyang et al., 2022) which are served as black-box language models through APIs[1]. The currently most advanced LLMs such as GPT-4 (OpenAI, 2023) is included in our test as well. We also experimented on open-source Vicuna-13B[2] and GPT-J (Wang and Komatsuzaki, 2021). The selection of black-box language models is not the primary focus of this study; rather, we seek to investigate IKA's universality.

### 4.3 Retrieval Methods for IKA

**Sparse Retrieval**

Sparse retrieval techniques are a class of traditional methods. Common sparse retrieval methods in-

---

[1]https://platform.openai.com/
[2]https://huggingface.co/TheBloke/wizard-vicuna-13B-HF

| Model | #Parameters | Open Source |
|-------|-------------|-------------|
| GPT-2 (2019) | 1.5B | ✓ |
| GPT-J (2021) | 6B | ✓ |
| Vicuna(2023) | 13B | ✓ |
| GPT-3.5(2023) | 175B | ✗ |
| GPT-4 (2023) | *unknown* | ✗ |

Table 1: A wide variety of large language models with parameters ranging from 1.5B to 175B are chosen in our experiments.

clude **TF-IDF** and **BM25** (Robertson and Walker, 1994). By arranging the documents into an inverted index, where each different word has an inverted list that contains information about the articles it appears in, this type of retrieval model can swiftly process queries. However, they often fall short of matching related words (Metzler and Croft, 2005) that are similar in meaning. Recent works have proposed to use contextualized representation to leverage the power of deep learning language models (Gao et al., 2021).

**Dense Retrieval**

In dense retrieval, queries and documents are often directly encoded by large neural models into single or multiple vectors, which allow for efficient *dense* retrieval based on simple similarity function such as dot-product or ANN. These vectors compress high dimensional semantic information.

Four types of dense retrieval are implemented in our experiments, the first three are bi-directional contextual language models (BERT (Devlin et al., 2018), RoBERTa (Liu et al., 2019), BART (Lewis et al., 2019)) using [CLS] embedding as the final input representation. The same encoder is used for both query and document. We also implement TCT-ColBERT (Wang et al., 2022) using index service provided by Pyserini[3] and FAISS[4] libraries.

**Dense-Sparse Hybrid**

The hybrid method can capture both lexical similarity and high-level semantic relevance. We further show that combining dense retrieval with sparse retrieval can improve performance. Following Lin and Ma (2021), TCT-ColBERT and Uni-COIL are chosen as dense retriever and sparse retriever in our hybrid setting. Through linear combination of scores, we adjust them to have the same order of magnitude.

[3]https://github.com/castorini/pyserini/
[4]https://github.com/facebookresearch/faiss

**Diverse Example Retrieval**

DER is implemented as described in 3.3. Note that the initial example set can be retrieved by any method, for simplicity we choose dense and hybrid retrieval described above. For distance measure, we choose RoBERTa (Liu et al., 2019) as the text encoder to obtain the dense vector representation and calculate the cosine distance between examples. The embeddings produced by RoBERTa capture semantic similarities between words and sentences, which has been proved to be beneficial for information retrieval tasks.

### 4.4 Baselines

We compare following strong baselines to demonstrate the efficacy of our framework. **RHO** (Ji et al., 2022) fine-tunes BART (Lewis et al., 2019) and GPT-2 (Radford et al., 2019), introduces local and global knowledge grounding techniques with an ad hoc re-ranking procedure. They report SOTA results on **OpendialKG** dataset. Other strong baselines are Sarkar et al. (2022) and Kang et al. (2022).

### 4.5 Evaluation Metrics

**General Generation Evaluation**

We conduct automated evaluation to measure the general generation quality and knowledge faithfulness of responses. Traditional word-overlap based metrics such as **BLEU**(Papineni et al., 2002) and **ROGUE-L**(Lin, 2004) are adopted. Meanwhile, some pre-trained models also exhibit promising results in term of evaluating generation informativeness, fluency and relevance, one among them is **BARTScore**(Yuan et al., 2021), which formulates evaluating text as a generation task and achieves better performance than compared metrics.

**Faithful Generation Evaluation**

For evaluating faithfulness, we consider **Entity Coverage** and **FeQA**(Durmus et al., 2020). Ji et al. (2022) argue that entities in generated responses should be covered by ones appear in dialog history and knowledge. Based on the entity collection of Freebase(Bollacker et al., 2008), we use exact match to extract named entities and calculate **Entity Precision, Recall and F1** score. In prior works (Ji et al., 2022; Dziri et al., 2021), question answering based metrics for faithfulness like **FeQA** are applied. FeQA first generates QA pairs from summary and then extracts answers from the document. We concatenate dialog history and knowledge as

| Model | BLEU4↑ | ROUGE-L↑ | BS†↑ | Entity Coverage%↑ | | | FeQA↑ | Avg Len |
|---|---|---|---|---|---|---|---|---|
| | | | | Pre | Recall | F1 | | |
| GPT-2 0-shot | 28.51 | 23.44 | 4.01 | 80.67 | 40.40 | 53.84 | 41.65 | 11.88 |
| GPT-J 0-shot | 39.37 | 29.99 | 4.02 | 87.70 | 36.28 | 51.33 | 44.27 | 13.29 |
| Vicuna 0-shot | 37.65 | 28.60 | 5.02 | 59.86 | 56.41 | 58.09 | 24.04 | 26.10 |
| GPT-3.5 0-shot | 46.38 | 34.86 | 5.37 | 76.69 | 69.74 | 73.05 | 43.74 | 16.12 |
| NPH* (2021) | 10.41* | 29.93* | - | 95.61* | 33.39* | 53.96* | - | - |
| RHO (2022) | 39.62 | 37.37 | 4.89 | 81.13 | 47.57 | 59.97 | **44.97** | 9.76 |
| IKA+GPT-2 | 27.14 | 20.92 | 3.65 | 53.15 | 27.19 | 35.98 | 23.76 | 12.36 |
| IKA+GPT-J | 40.45 | 31.00 | 5.04 | 57.86 | 40.70 | 47.79 | 31.02 | 16.45 |
| IKA+Vicuna | 50.28 | 41.14 | 7.03 | 65.27 | 58.63 | 61.77 | 31.43 | 17.63 |
| IKA+GPT-3.5 | **58.11** | **47.09** | **10.90** | 79.12 | **77.41** | **78.26** | 44.67 | 17.08 |
| IKA+GPT-4 | 49.44 | 39.45 | 7.12 | 73.33 | 77.73 | 75.47 | 37.99 | 21.07 |
| Gold Res. | 100 | 100 | 47.38 | 100 | 100 | 100 | 49.73 | 17.75 |

Table 2: Experimental results on **OpendialKG** under different settings. For experiments with IKA, we use hybrid retrieval described in 4.3, the number of examples is set to 3. *The results of NPH are reported in (Ji et al., 2022); the results of RHO(Ji et al., 2022) are calculated from their publicly available generated results[5]. †BS stands for BARTScore; Gold Res. stands for ground truth response.

the document and response as the summary to measure whether the generated response is faithful to the original context(Dziri et al., 2021).

## 5 Results and Analysis

The experimental results on **OpendialKG** are summarized in Table 2 and Table 3. The performance of IKA is compared to previous training-based methods and zero-shot baselines.

### 5.1 Main Results

**IKA greatly improves knowledge grounded dialogue generation quality.** Table 2 summarizes the results of experiments on previous SOTA model and LLMs. The top four rows report the zero-shot performance of different LLMs, and GPT-3.5 achieves the best generation performance among them, 46.38 in BLEU4, 34.86 in ROUGE-L and 5.37 in BARTScore, which is in line with recent studies (Hendy et al., 2023). We also report the average length of generated outputs. Vicuna generates the longest 26.10 words under 0-shot setting, which results in the lowest FeQA score. It seems that Vicuna has the tendency to produce redundant nonsense. However when we combine Vicuna with IKA, the average length of generated response significantly decreases while all evaluation metrics increase. Similar improvements are observed across different backbone LLMs except for GPT-2. One possible explanation may be the lack of ICL ability for smaller language models like GPT-2.

Row 5 and Row 6 in Table 2 correspond to previous SOTA. The performance of our framework when paired with various LLMs is presented in the next five lines. Our framework outperforms SOTA method (Ji et al., 2022) on general generation metrics and faithful generation metrics, with a significant rise of **46.67%** in BLEU4, **26.01%** in ROUGE-L, **122.90%** in BARTScore and **30.50%** in Entity Coverage F1. In order to keep the results comparable, we filter and split our training and test set so that they are both subsets of Ji et al. (2022). We only evaluate on the common test samples in Table 2.

**The retrieval method is vital to the performance of IKA.** In Table 3, we report the results of several types of retrieval methods. Here we fix the retrieved example number to 3. The first row of Table 3 is random baseline. Although our reference dataset has been filtered in advance which means even random sampling can retrieve relatively high quality examples, the improvement obtained is still insignificant compared to the zero-shot baseline reported in Table 2. If we choose a proper retrieval method such as BM-25 (Robertson and Walker, 1994) or TCT-ColBERT (Wang et al., 2022) the performance will boost. The hybrid retrieval method achieves the highest performance in BLEU4, ROUGE-L and BARTScore, with the rise of **18.43%**, **27.75%** and **69.78%** respectively compared to random baseline.

**Our diverse strategy improves models' faith-**

| Retrieval Methods | | BLEU4↑ | ROUGE-L↑ | BS↑ | Entity Coverage%↑ | | | FeQA↑ |
| | | | | | Pre. | Recall | F1 | |
|---|---|---|---|---|---|---|---|---|
| Sparse | Random | 47.86 | 36.04 | 5.99 | 80.46 | 71.44 | 75.68 | 43.01 |
| | BM-25 | 51.98 | 41.20 | 7.59 | 80.46 | 73.40 | 76.77 | 43.71 |
| | Uni-COIL | 56.60 | 45.97 | 9.97 | 80.85 | 75.06 | 77.85 | 44.15 |
| Dense | BERT | 55.63 | 39.42 | 7.14 | 81.09 | 75.36 | 77.79 | 43.94 |
| | RoBERTa | 46.42 | 35.74 | 5.66 | 82.03 | 69.27 | 75.11 | 47.65 |
| | BART | 48.49 | 37.03 | 6.20 | 81.13 | 70.20 | 75.27 | 46.32 |
| | TCT-CBT | 50.67 | 39.06 | 6.67 | **82.03** | 73.15 | 77.33 | 44.12 |
| Hybrid | Hybrid | **56.68** | **46.14** | **10.17** | 81.32 | **75.53** | 78.32 | 44.67 |
| Diverse | U-C div | 56.17 | 45.61 | 9.99 | 81.14 | 75.58 | 78.26 | 44.80 |
| | Hybrid div | 52.89 | 42.34 | 7.85 | **84.47** | 74.21 | **79.01** | **48.08** |

Table 3: Comparison of different retrieval methods. Only top-3 retrieval results are used in the final prompts. Random sampling serves as the baseline. Among all retrieval methods the hybrid method achieves the highest scores on general quality metrics, whereas DER further improves faithfulness metrics while sacrificing generation metrics.

| #EX | Triplet | | Text | |
| | BS | EC-F1 | BS | EC-F1 |
|---|---|---|---|---|
| 1 | 6.48 | 74.97 | 5.99 | 74.23 |
| 3 | 10.90 | 77.16 | 9.99 | 75.91 |
| 5 | 13.16 | **77.62** | 12.52 | 77.50 |
| 10 | **15.12** | 77.25 | **14.38** | **77.82** |

Table 4: Comparison of knowledge representation.

**fulness generation ability.** In Section 3.3 we propose the diverse strategy for ICL example selection. Since it involves the trade-off between relevance and diversity controlled by the hyper-parameter $w$, additional hyper-parameter search is required in practice. Due to resource constraint we empirically set a value for $w$ based on performance on dev set, and it is able to achieve the highest Entity Coverage F1 at **79.01** and FeQA at **48.08** with a slight drop on general quality metrics. A HP search for coefficients may further boost the performance of DER.

## 5.2 Analysis

In this section, we give some analysis on different aspects of our IKA framework.

### On the High Scores of Generation Metrics

The BLEU and ROUGE scores in Table 2 and 3 may seem extraordinary high for a text generation task. The task of knowledge grounded dialog generation focuses on the usage of knowledge. Its task structure inherently leads to relatively fixed outputs, as the majority of the information is derived from the input knowledge. Having a high BLEU/ROUGE score for such a task is quite rea-

sonable, since these metrics only measure word-overlapping in discrete string space. In summation, metrics to explicitly measure how well models or frameworks can generate faithfully are needed.

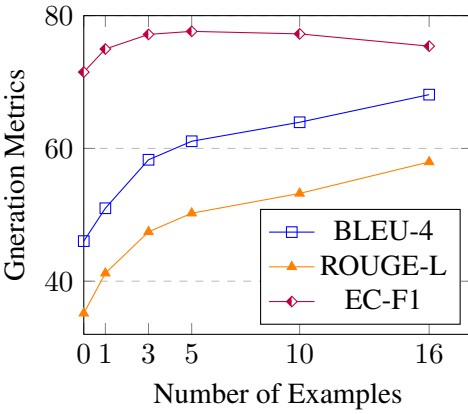

Figure 4: Generation metrics for IKA under different number of examples.

### On the Impact of the Number of Examples

According to Brown et al. (2020), the number of demonstrations is an important factor in ICL. To test the scalability of IKA and also keep the constructed prompt within context window size, we experiment the number of examples from 0 to 16. Figure 5.2 illustrates the results. As the number grows, IKA generates more high quality response; but the EC-F1 metric rises and then falls, which means there might be a trade-off between faithfulness and the generation quality. The more examples provided, the better LLMs can mimic conversation style which contributes to the improvements in evaluation metrics.

| Ret. | #EX | BLEU4 | ROUGE-L | BS |
|------|-----|-------|---------|-----|
| None | 0 | 13.99 | 9.39 | 2.92 |
| Rand. | 3 | 58.09 | 58.62 | 23.15 |
| BM25 | 3 | 60.35 | 61.09 | 24.78 |

Table 5: Results of IKA+GPT3.5(text-davinci-003) on Wizard-of-Wikipedia(Dinan et al., 2018) unseen split results.

**On the Choice of Knowledge Representation**

We compare the performance of IKA given different forms of knowledge including knowledge graphs and **texts**. Results on OpendialKG are presented in Table 4. Surprisingly the knowledge triplet is almost always better no matter how many examples are presented. One possible reason is that the knowledge triplet is more suitable for example retrieval given their succinct triplet form, while textual knowledge may provide redundant information. Besides, the structured form is more compact which passes the most essential knowledge to LLMs. Results on Wizard-of-Wikipedia (Dinan et al., 2018) are summarized in Table 5, where only textual knowledge is provided. Under 3-example setup, we compare random retrieval and BM25 retrieval. On unseen test split of WoW, BLEU and ROUGE-L increase from 58.09 to 60.35 and 58.62 to 61.09 respectively, BARTScore increases from 23.15 to 24.78. These results demonstrate the universality of our framework across different forms and areas of knowledge.

## 6 Conclusion

In this paper, we explore the strategy of adopting in-context learning for knowledge grounded dialog generation. We propose a plug-and-play retrieval based framework **IKA** that can be easily integrated with LLMs. Our framework **significantly** outperforms strong baselines and previous SOTAs without additional parameters update. We explicitly model the dependency between ICL examples and design a diversification strategy that further improves performance. Experiments demonstrate the effectiveness and generalization of our framework. Our results also provide the evidence for ICL's potentials in knowledge-intensive tasks.

## Limitations

In this work, we do not study the knowledge selection process which is also important to knowledge grounded generation. Following previous works, we assume that the gold relevant knowledge is provided in target context. How LLMs integrate knowledge is also under-studied. Due to text window size constraints, we are unable to obtain results for more examples. Last, we do not perform human evaluation to analyze different types of faithfulness errors in our generated results. We leave these open questions for future work.

## Ethics Statement

Our work complies with the ACL Ethics Policy. All datasets and models are publicly accessible except for OpenAI's text-davinci-003 and GPT-4. We have not identified any significant ethical considerations associated with our work. We believe our findings can help reducing LLMs' hallucinations.

## Acknowledgement

We thank the anonymous reviewers for their helpful comments on this paper. This work was partially supported by National Key R&D Program of China (No. 2022YFC3600402) and National Social Science Foundation Project of China (21&ZD287). The corresponding author of this paper is Sujian Li.

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
