# OpenReview forum: "Exploring In-Context Learning for Knowledge Grounded Dialog Generation"
_EMNLP/2023/Conference — EMNLP 2023 Findings_

### Official Review · Reviewer_fq1q · 2023-07-31

**Soundness:** 2

**Excitement:**

3: Ambivalent: It has merits (e.g., it reports state-of-the-art results, the idea is nice), but there are key weaknesses (e.g., it describes incremental work), and it can significantly benefit from another round of revision. However, I won't object to accepting it if my co-reviewers champion it.

**Missing References:**

N/A

**Paper Topic And Main Contributions:**

This paper proposes to leverage in-context learning and retrieval approach to enhance large language models on knowledge grounded dialog generation. Specifically, their prompt includes instruction, in-context examples, dialog history and knowledge. To retrieve the in-context examples, they designed a diverse example retrieval (DER) approach. They worked a knowledge grounded dialog dataset OpendialKG that has annotated with knowledge graph and experimented with different retrieval strategies and LLMs. Their results demonstrated that their plug-and-play retrieval-based framework IKA can improve knowledge grounded dialogue generation quality and the retrieval approach is vital to the performance. Their diverse retrieval strategy doesn't perform better than the sparse and dense hybrid approach, but can improve model's faithfulness generation ability.

**Questions For The Authors:**

1. How do you calculate the distance between each example in DER?
2. While you approach leverage existing datasets with knowledge graph, how about the performance on the datasets without knowledge graph?


**Reasons To Accept:**

They apply in-context learning and different retrieval strategies for knowledge grounded dialog generation. And their experiments demonstrate the good performance on OpendialKG dataset.

**Reasons To Reject:**

Firstly, the main contribution of this paper is to leverage in-context learning (ICL) and retrieval for knowledge grounded dialog generation, but they don't have a innovated approach for both concepts. Secondly, their proposed diversified example retrieval strategy doesn't outperform over hybrid (sparse and dense) retrieval strategy overall. Lastly, they only experiment on one dataset OpendialKG which doesn't justify their approach can be generalized to other datasets or tasks.

**Reproducibility:**

3: Could reproduce the results with some difficulty. The settings of parameters are underspecified or subjectively determined; the training/evaluation data are not widely available.

**Reviewer Confidence:**

4: Quite sure. I tried to check the important points carefully. It's unlikely, though conceivable, that I missed something that should affect my ratings.

**Typos Grammar Style And Presentation Improvements:**

In table 2, the column AVG in the rightmost might be misleading as people might think it denotes the average performance. In line 301, you have the same notation K for relevant examples and knowledge graph.

---

> ### Author Rebuttal · Authors · 2023-08-28
>
> Thank you for the constructive review! We will fix the typo on line 301 and add clarification for the AVG column in Table 2 immediately.
>
> Regarding the novelty of our method, we do not regard the combination of LLM and retrieval is trivial. This is because while the training process of LLMs does not encompass retrieval, the inference of LLMs heavily leans on the retrieval module, especially in the context of knowledge grounded dialog generation. We have clarified this distinction in the paper, highlighting how our combined approach offers a new perspective and SOTA results in this context.
>
> Answer to Q1: How to calculate distance in DER?
>
> We use RoBERTa as a text encoder to obtain the vector representation of in-context-learning samples, then we use consine distance to calculate the distance between each example. The embeddings produced by RoBERTa capture semantic similarities between words and sentences, which has been proved to be beneficial for information retrieval tasks.
>
> Answer to Q2: How about performance on dataset without knowledge graph?
>
> We did conduct experiments that did not use knowledge graphs. We use the textual form of knowledge, originally provided in Opendialkg dataset. The results of these experiments are reported in Table4. We plan to emphasize in revised version that our experiments are not only performed on the knowledge graphs.
>
> The reasons we chose OpendialKG over Wizard-of-Wikipedia are as follows:
>
> 1. The main difference between OpendialKG and Wizard-of-Wikipedia is that OpendialKG provides two forms of knowledge representation, both triplet and texual. We experimented on both forms, the results are presented in Table4.
> 2. The quality of dataset. The Wizard-of-Wikipedia training set contains 18340 dialogs, but some of the samples in test set also appear in training set so there is data leak. In addition, about 50% of wizards' responses do not utilize any knowledge, even if they do, many of the referenced knowledge are imcomplete segments from wikipedia. According to [1], 60%~70% of WoW responses are hallucinated or partially hallucinated. Thus we doubt the choice of WoW as a fair evaluation.
>
> Despite the problems mentioned above, we still conducted additional experiments on WoW. Below are our results on WoW unseen test set. The first column stands for different retrieval methods, the scond column stands for different number of in-context-learning examples, the last three columns are BLEU, ROUGE-L and BARTscores.
>
> | Retrieval | #EX  |    BLEU    |  ROUGE-L   | BARTScore  |
> | :-------: | :--: | :--------: | :--------: | :--------: |
> |   None    |  0   |   0.1399   |   0.0939   |   2.9168   |
> |  Random   |  3   |   0.5809   |   0.5862   |   23.146   |
> |   BM25    |  3   | **0.6035** | **0.6109** | **24.782** |
>
> We filter out wizards' responses that do not provide knowledge in both training set and test set. For comparison, we also choose text-davinci-003 as the base LLM. Under 0-shot setting, the BLEU/ROUGE-L/BARTScore are 0.1399/0.0939/2.9168, which demonstrates the great potential of in-context-learning in KGD. Under 3-example setting, we compare random retrieval and BM25 top-k retrieval. On unseen test split of WoW, BLEU and ROUGE-L increase from 0.5809 to 0.6035 and 0.5862 to 0.6109 respectively. BARTScore increases from 23.146 to 24.782. Besides, we will append case study in the revised version.
>
> [1] Dziri, Nouha, et al. "On the origin of hallucinations in conversational models: Is it the datasets or the models?." *arXiv preprint arXiv:2204.07931* (2022).

---

### Official Review · Reviewer_WFsR · 2023-08-01

**Typos Grammar Style And Presentation Improvements:** n/a
**Soundness:** 3

**Excitement:**

4: Strong: This paper deepens the understanding of some phenomenon or lowers the barriers to an existing research direction.

**Missing References:**

n/a

**Paper Topic And Main Contributions:**

The primary focus of this paper is on constructing a knowledge-grounded dialogue (KGD) system. While existing LLM-based KGD systems lack full control over generated content and may sometimes experience hallucinations, this paper presents an innovative approach. The author tackles the knowledge-based dialogue generation problem by transforming it into an example retrieval challenge using off-the-shelf large language models (LLMs) and in-context learning.

Two significant contributions stand out: firstly, this work pioneers the modeling of KGD within the in-context learning framework. Secondly, the authors propose a retrieval-based framework that outperforms the previous state-of-the-art (SOTA) by a substantial margin. By leveraging these novel advancements, the paper showcases a promising path for more effective and controlled knowledge-grounded dialogue generation.

**Questions For The Authors:**

1. Is Table 3 comparable to Table 2?  Aside of different retrieval methods, what is the LLM used for Table 3?
2. When retrieving examples for in-context learning, what is the source?  Does it only retrieve from the training set samples or the whole set?

**Reasons To Accept:**

1. The integration of in-context learning into KGD, and the transformation of the knowledge encoding and dialogue generation problem into an example retrieval challenge, represent a novel and intriguing approach. The potential implications of this work for future research in this domain are significant and valuable.

2. The IKA framework proposed in this study has demonstrated a remarkable improvement in the state-of-the-art performance, achieving an impressive 46.67% increase in BLEU4 and 26.01% in ROUGE-L scores. While part of these gains can be attributed to the utilization of GPT 3.5, Table 2 highlights IKA's substantial enhancement of GPT 3.5 in the KGD task. Extensive ablation experiments underscore the importance of retrieval quality for performance, and the proposed diverse example retrieval strategy stands out with its highest entity coverage and FeQA scores. These findings provide compelling evidence of the efficacy of the IKA framework in pushing the boundaries of performance in knowledge-grounded dialogue generation.

**Reasons To Reject:**

The remarkable improvements achieved can largely be attributed to the use of GPT 3.5. In Table 2, IKA does not yield significant additional gains when applied to GPT2 and GPT-J. Its benefits become evident only when combined with Vicuna and GPT 3.5. Furthermore, the comparison in Row 4 of Table 2 highlights the substantial superiority of GPT 3.5 0-shot over the previous SOTA. An educated assumption suggests that GPT 3.5 likely possesses a wealth of external knowledge beyond the training data, making it an unequal comparison with the previous SOTA and the proposed methods. These observations collectively underscore the heavy reliance of the proposed method on a powerful LLM like GPT 3.5.

**Reproducibility:**

3: Could reproduce the results with some difficulty. The settings of parameters are underspecified or subjectively determined; the training/evaluation data are not widely available.

**Reviewer Confidence:**

4: Quite sure. I tried to check the important points carefully. It's unlikely, though conceivable, that I missed something that should affect my ratings.

---

> ### Author Rebuttal · Authors · 2023-08-28
>
> We greatly appreciate your recognition of our work and valuable feedback!
>
> In response to your concern, we admit our reliance on powerful LLMs, while models like GPT-2 or GPT-J are usually used in a finetuned manner. The difference may also come from the training corpus and parameter size of LLMs, which is hard to control in our setting. Our method does have some scope for use, we will clarify this point in our revised version. Thank you for helping us refine our work and clarify the role of GPT-3.5 in our proposed method's success.
>
> Answer to Q1: Is Table 3 and Table 2 comparable? What LLM?
>
> Yes, the results in Table 3 is comparable to Table 2. We used text-davinci-003 for Table 3.
>
> Answer to Q2:  What is the source for retrieval?
>
> We retrieve examples from training split rather than the whole set. Training set consists of held-out dialogs that have no overlap with the test set.
>
> We appreciate the insights provided and are committed to addressing these issues in our revised paper.

---

### Official Review · Reviewer_MTiQ · 2023-08-04

**Soundness:** 3

**Excitement:**

2: Mediocre: This paper makes marginal contributions (vs non-contemporaneous work), so I would rather not see it in the conference.

**Paper Topic And Main Contributions:**

This paper proposes a plug-and-play retrieval method to improve in-context learning of knowledge-grounded dialogue generation. It retrieves both diverse and relevant demonstrations in a Markov process manner.

**Questions For The Authors:**

1. Why simple linear combination of dense retrieval score and sparse retrieval score works? It seems their scores' distribution and range are different.
2. The too-high BLEU/ROUGE score is unnormal -- BLEU-4 ≈ 50 is crazy. It usually means the dataset may not be capable of reflecting the practical performance of methods, or the task has been well solved.

**Reasons To Accept:**

1. The proposed method is intuitive, including retrieving both diverse and relevant demonstrations and using both dense and sparse retrieval methods.
2. The experiment indicates the method outperforms other methods in the OpendialKG dataset.

**Reasons To Reject:**

1. Although the method improves several metrics, few insights and analyses are presented. The research needs more analysis to prove the effectiveness, such as why the method improves in-context learning, why different retrieval method has different performances, which demonstrations are preferred by LLMs, and showing cases to help compare and understand why the retrieved demonstrations improve performance.
2. The retrieval framework is not designed for knowledge graph retrieval but is only evaluated on OpendialKG. Thus, knowledge-grounded dialogue generation datasets like Wizard-of-Wiki, grounding on documents, are worth evaluating.

**Reproducibility:**

4: Could mostly reproduce the results, but there may be some variation because of sample variance or minor variations in their interpretation of the protocol or method.

**Reviewer Confidence:**

4: Quite sure. I tried to check the important points carefully. It's unlikely, though conceivable, that I missed something that should affect my ratings.

---

> ### Author Rebuttal · Authors · 2023-08-28
>
> We appreciate your feedback and insights on our paper.
>
> We acknowledge the importance of providing deeper insights and analyses to substantiate the effectiveness of our method.  Due to page limit, we were unable to present our full analysis of reasons behind the observed improvements and the influence of different retrieval methods. We will enhance the analysis section in revised version of our paper.
>
> The reasons we chose OpendialKG over Wizard-of-Wikipedia are as follows:
>
> 1. The main difference between OpendialKG and Wizard-of-Wikipedia is that OpendialKG provides two forms of knowledge representation, both triplet and texual. We experimented on both forms, the results are presented in Table4.
> 2. The quality of dataset. The Wizard-of-Wikipedia training set contains 18340 dialogs, but some of the samples in test set also appear in training set so there is data leak. In addition, about 50% of wizards' responses do not utilize any knowledge, even if they do, many of the referenced knowledge are imcomplete segments from wikipedia. According to [1], 60%~70% of WoW responses are hallucinated or partially hallucinated. Thus we doubt the choice of WoW as a fair evaluation.
>
> Despite the problems mentioned above, we conducted additional experiments on WoW. Below are our results on WoW unseen test set. The first column stands for different retrieval methods, the scond column stands for different number of in-context-learning examples, the last two columns are BLEU and ROUGE-L scores.
>
> | Retrieval | #EX  |    BLEU    |  ROUGE-L   | BARTScore  |
> | :-------: | :--: | :--------: | :--------: | :--------: |
> |   None    |  0   |   0.1399   |   0.0939   |   2.9168   |
> |  Random   |  3   |   0.5809   |   0.5862   |   23.146   |
> |   BM25    |  3   | **0.6035** | **0.6109** | **24.782** |
>
> We filter out wizards' responses that do not provide knowledge in both training set and test set.  All ICL examples are retrieved from the training set. For comparison, we also choose text-davinci-003 as the base LLM. Under 0-shot setting, the BLEU/ROUGE-L/BARTScore are 0.1399/0.0939/2.9168, which demonstrates the great potential of in-context-learning in KGD. Under 3-example setting, we compare random retrieval and BM25 top-k retrieval. On unseen test split of WoW, BLEU and ROUGE-L increase from 0.5809 to 0.6035 and 0.5862 to 0.6109 respectively. BARTScore increases from 23.146 to 24.782. We will append case study in the revised version.
>
> Answer to Q1: Why simple linear combination works?
>
> About the hybrid retrieval, we basicly follow the setting reported in [2]. The scores range for dense retrieval and sparse retrieval is different, but they are on the same order of magnitude. The retrieval score is calculated as alpha \* sparse_score + beta \* dense_score, where alpha and beta are 0.5 and 1 respectively. We choose hyper-parameters according to performance on validation set. More experimental details will be added in appendix, including the HP search process and model choice.
>
> Answer to Q2: Too-high score means the dataset is uncapable?
>
> The task of knowledge grounded dialog generation focuses on the usage of knowledge. This task structure inherently leads to relatively fixed outputs, as the majority of the information is derived from the input knowledge. Unlike open domain dialog generation that aims for diversity, task of this nature prioritize accuracy in conveying the correct knowledge. Having a high BLEU/ROUGE score for such a task is quite reasonable, since these metrics only measure word-overlapping in discrete string space.
>
> We provide other evaluation metrics such as BARTScore and FeQA which leverage more pre-trained parameters and more robust compared to BLEU/ROUGE. Besides, the faithfulness of dialog generation is another important aspect we consider, which can not be reflected by BLEU or ROUGE scores. In terms of these metrics, KGD task is far from solved.
>
> Again, thank you for your constructive feedback, we are dedicated to addressing these issues!
>
> [1] Dziri, Nouha, et al. "On the origin of hallucinations in conversational models: Is it the datasets or the models?." *arXiv preprint arXiv:2204.07931* (2022).
>
> [2]Lin, Jimmy, and Xueguang Ma. "A few brief notes on deepimpact, coil, and a conceptual framework for information retrieval techniques." *arXiv preprint arXiv:2106.14807* (2021).

---

### Meta-Review · Area_Chair_7JJK · 2023-10-04

**Recommendation:** 3

**Metareview:**

Plug-and-play retrieval + in-context learning-based dialog generation for knowledge grounding. While the method is intuitive, the paper lacks a proper demonstration of how each component of the model contributes to a high performance.

---

### Decision · Program_Chairs · 2023-10-07

**Decision:**

Accept-Findings

**Comment:**

Plug-and-play retrieval + in-context learning-based dialog generation for knowledge grounding. While the method is intuitive, the paper lacks a proper demonstration of how each component of the model contributes to a high performance.